# Position: Hyperbolic Embeddings Are Essential for Health Knowledge Graphs in LLMs and Vector Databases

## Abstract

This position paper contends that hyperbolic embeddings must become a standard for modeling and retrieving hierarchical health knowledge graphs (HKGs) within large language models (LLMs) and their supporting vector databases. While Euclidean or spherical embeddings remain prevalent in biomedical retrieval systems, these geometries cannot adequately capture the deep ontological hierarchies, small-world connections, and rich relational patterns inherent in medical data. By contrast, hyperbolic embeddings exploit negatively curved spaces such as the Poincaré ball to compress hierarchical information with minimal distortion, paving the way for more interpretable retrieval, advanced question answering, and robust clinical decision support. This paper details how negative curvature addresses common bottlenecks in Euclidean-based solutions and calls on the healthcare and ML communities to adopt hyperbolic geometry as a core component of next-generation health informatics pipelines. We present both theoretical underpinnings and practical implementation strategies, supplemented by four in-depth appendices that cover mathematical proofs, comprehensive literature overviews, experiment design frameworks, and real-world policy considerations. Despite engineering and organizational hurdles, we argue that hyperbolic embeddings offer compelling benefits and should be the default choice for hierarchical HKGs in LLM-driven ecosystems.

Keywords: Hyperbolic embedding, Health knowledge graphs, Large language model, Vector database

[1] Anonymous Institution, Anonymous City, Anonymous Region, Anonymous Country. Correspondence to: Anonymous Author <anon.email@domain.com>.

Preliminary work. Under review by the International Conference on Machine Learning (ICML). Do not distribute.

## 1. Introduction

*Position Statement: Hyperbolic embeddings should become a standard for encoding and retrieving hierarchical health knowledge graphs (HKGs) within large language models (LLMs) and their supporting vector databases.*

Modern health informatics integrates diverse data sources such as disease ontologies, molecular interactions, and patient records (Callahan et al., 2013; Mungall et al., 2017). These large-scale HKGs often exhibit deep hierarchical layers (e.g., multilevel ICD or SNOMED structures) alongside small-world connections (e.g., cross-links between related diagnoses). Traditional Euclidean or spherical embeddings, however, struggle to capture such tree-like depth without high-dimensional overhead, leading to suboptimal retrieval performance in LLM-based systems (Devlin et al., 2019; Lee et al., 2020; Gu et al., 2021; Singhal et al., 2023).

We argue that *hyperbolic embeddings* provide a more natural fit, thanks to their negative curvature property that aligns with branching ontologies. Prior research shows that hyperbolic spaces not only reduce distortion but also embed complex hierarchies in fewer dimensions (Nickel & Kiela, 2017; 2018; Sala et al., 2018). This leads to more interpretable boundaries among disease subgroups and improved retrieval fidelity—crucial for clinical trust and decision support. By adopting a *Hyperbolic HKG Pipeline* (see Figure 1), we can integrate curvature-tuned training, specialized vector database indexing, and LLM-driven queries into a cohesive system that better reflects real-world healthcare complexity.

This paper lays out the mathematical rationale for hyperbolic geometry in health informatics, detailing how negative curvature combats the exponential blow-up that plagues Euclidean embeddings in deeply layered structures. We then highlight practical considerations—rewriting vector databases, fine-tuning curvature, and managing dynamic ontology updates—underscoring the need for interdisciplinary collaboration among clinicians, informaticians, and AI researchers.

In the appendices, we provide technical proofs (*Appendix A*), a thorough literature survey (*Appendix B*), a proposed

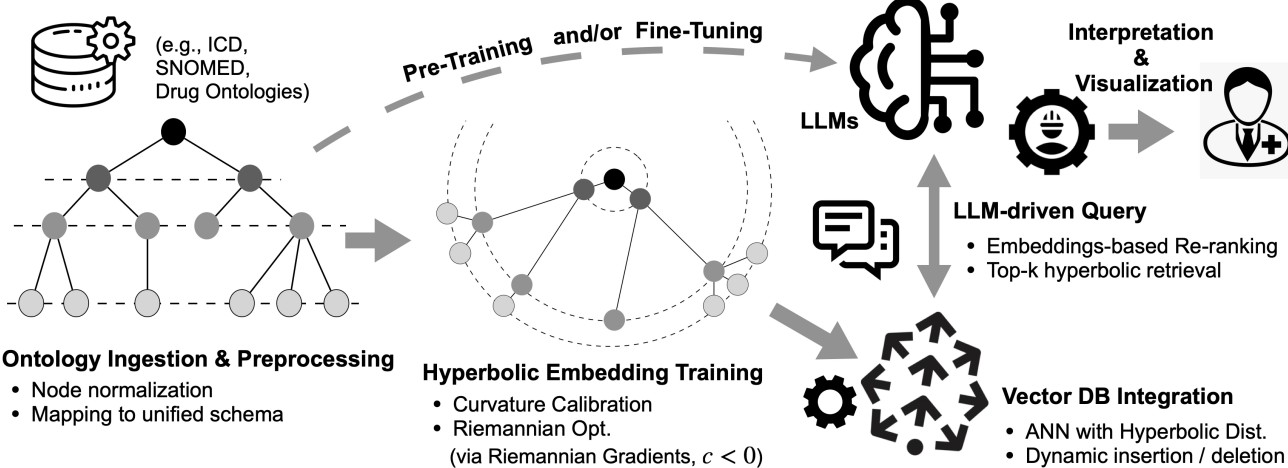

Figure 1. *Hyperbolic HKG Pipeline* for integrating hyperbolic embeddings into health knowledge graphs. The pipeline encompasses: (1) **Ontology Ingestion & Preprocessing** for node normalization and unified-schema mapping; (2) **Hyperbolic Embedding Training** with curvature calibration and Riemannian optimization (via negative curvature, $c < 0$); (3) **Vector DB Integration** supporting approximate nearest-neighbor queries under hyperbolic distance and dynamic insertion/deletion; (4) **LLM-driven Query** enabling embeddings-based re-ranking and top-$k$ hyperbolic retrieval; and (5) **Interpretation & Visualization** modules for clinical end-users. This unified framework highlights how hierarchical fidelity and negative curvature can be harnessed to build robust, scalable, and interpretable healthcare systems.

experimental design (*Appendix C*), and deployment roadmaps alongside policy insights (*Appendix D*). Our goal is to show that hyperbolic embeddings are not an esoteric choice but a *practical and necessary* strategy to build interpretable, hierarchically faithful retrieval frameworks for ever-growing healthcare data.

## 2. Background and Related Work

### 2.1. Health Knowledge Graphs

The concept of HKGs stems from the need to integrate heterogeneous health data – from biomedical ontologies (e.g., SNOMED CT, ICD, UMLS) to de-identified clinical and genomic records – into a cohesive, queryable framework. HKGs connect entities such as diseases, treatments, genes, and patient demographics (Callahan et al., 2013; Mungall et al., 2017), with edges capturing causal, taxonomic, or associative relationships. As these graphs expand to millions of nodes and edges, capturing both deep hierarchical relationships (e.g., disease subtypes) and small-world effects (e.g., multiple cross-links among related conditions) becomes increasingly complex.

### 2.2. LLMs in Health Informatics

LLMs, including BioBERT (Lee et al., 2020) and PubMedBERT (Gu et al., 2021), have raised the bar on tasks like medical entity extraction and relation classification. More general-purpose models like GPT-4 are showing

promise in sophisticated tasks such as medical question answering and summarizing clinical guidelines (Singhal et al., 2023). However, LLMs often rely on vector retrieval layers to serve up relevant knowledge. Most off-the-shelf vector databases assume Euclidean (or sometimes spherical) embeddings, limiting their ability to encode the nuanced hierarchies and domain-specific complexities of health data.

### 2.3. Vector Databases and Non-Euclidean Embeddings

High-performance vector databases (e.g., FAISS or HNSW-based solutions) provide the backbone for large-scale similarity searches (Johnson et al., 2021). While these systems have proven extremely efficient in Euclidean or cosine-based vector spaces, they do not readily incorporate alternative distance metrics that might better represent tree-like structures. Researchers have begun exploring hyperbolic embeddings (Nickel & Kiela, 2017; 2018; Sala et al., 2018; Chami et al., 2020) in contexts like link prediction and hierarchical taxonomy encoding, but widespread adoption in health informatics pipelines remains limited.

### 2.4. Hyperbolic Geometry in Machine Learning

Hyperbolic embeddings exploit negatively curved spaces to encode hierarchical relationships more naturally than their Euclidean counterparts. Pioneering works have demonstrated that the Poincaré ball model preserves large hierarchical ontologies in low dimensions (Nickel & Kiela,

2017), and subsequent research has extended these ideas with hyperbolic graph neural networks (Chami et al., 2019), hyperbolic word embeddings (Tifrea et al., 2019), and hyperbolic approaches for large-scale knowledge graphs (Monath et al., 2019). Their theoretical strength lies in the exponential growth of volume with respect to radius, aligning well with how nodes proliferate at each level of a taxonomy.

### 2.5. Gaps in Adoption for Health Informatics

Despite evidence that hyperbolic embeddings can reduce distortion and dimensional requirements, most clinical knowledge retrieval systems remain anchored to Euclidean-based index structures (Nickel et al., 2015). One major barrier is the perceived complexity of implementing hyperbolic distance metrics and approximate nearest neighbor (ANN) searches. Another is the inertia of existing workflows and standards in hospital settings. Consequently, advanced geometry for better hierarchical representation has not yet gained the traction it deserves in real-world healthcare systems.

Moreover, Appendix B provides a comprehensive review and classification of relevant literature.

## 3. Why Hyperbolic Embeddings? Significance and Evidence

Hyperbolic embeddings offer a principled and powerful approach for representing HKGs that exhibit deep hierarchical layers, small-world phenomena, and complex relational structures. Building upon the motivations in the Introduction, this section not only summarizes the theoretical and empirical justifications for negatively curved spaces, such as the Poincaré ball, but also introduces a *systematic implementation framework* that highlights how hyperbolic embeddings can be integrated into real-world LLM-driven health information systems. This proposed framework is one of our main contributions, offering a step-by-step methodology for practitioners to adopt hyperbolic geometry in clinical or biomedical pipelines. We further highlight references to our appendices, which provide additional mathematical details (*Appendix A*), extended literature insights (*Appendix B*), experimental design outlines (*Appendix C*), and practical policy considerations (*Appendix D*).

### 3.1. Aligning Negative Curvature with Hierarchical Health Data

Health ontologies and classification schemes typically manifest as multi-layered, tree-like or DAG-based structures, with entity depth often reaching six or more levels in resources like SNOMED CT or ICD (Callahan et al., 2013;

Mungall et al., 2017). Hyperbolic geometry naturally accommodates such branching because distances expand exponentially as one moves away from the origin. Formally, for a $d$-dimensional Poincaré ball

$$\mathbb{D}^d = \{\mathbf{x} \in \mathbb{R}^d : \|\mathbf{x}\| < 1\}$$

the distance between two points $\mathbf{u}, \mathbf{v}$ is

$$d_{\mathbb{D}}(\mathbf{u}, \mathbf{v}) = \operatorname{arcosh}\left(1 + 2\frac{\|\mathbf{u} - \mathbf{v}\|^2}{(1 - \|\mathbf{u}\|^2)(1 - \|\mathbf{v}\|^2)}\right) \quad (1)$$

and $d_{\mathbb{D}}(\mathbf{u}, \mathbf{v})$ grows rapidly as $\|\mathbf{u}\| \to 1$. This property is crucial in embedding hierarchical structures: top-level concepts map near the center, while more specialized or granular nodes populate regions closer to the boundary (Nickel & Kiela, 2017; 2018). As detailed in *Appendix A*, negative curvature fosters compact tree embeddings, preventing the dimensional explosion that Euclidean spaces often require for comparable fidelity.

### 3.2. Low-Dimensional Fidelity and Distortion Boundaries

A defining advantage of hyperbolic geometry is its ability to maintain low distortion across multiple levels of a hierarchy without resorting to high-dimensional embeddings. While Euclidean approaches must frequently increase dimension to capture deep ontological nuance, hyperbolic spaces distribute nodes efficiently along radial geodesics. Theorem 3.1 below, which builds upon the foundational work (Nickel & Kiela, 2017; Sala et al., 2018) and is extended in *Appendix A*, formalizes this fundamental principle:

**Theorem 3.1** (Simplified Distortion Bound). *Let $\mathcal{T}$ be a tree with branching factor $b$ and height $h$. Embedding $\mathcal{T}$ into a $d$-dimensional Poincaré ball $\mathbb{D}^d$ yields a maximum pairwise distortion $\delta$ that grows only logarithmically in $(b, h)$, whereas an equivalent Euclidean embedding of $\mathcal{T}$, for comparable distortion, typically grows in dimension at least linear in $h$.*

By keeping distortion in check as tree depth increases, hyperbolic embeddings reduce computational overhead in downstream tasks such as link prediction, subgraph detection, or semantic retrieval. In large-scale health systems, where disease categories often nest six or more layers deep, the capacity to embed thousands of node types in a compact space can yield substantial efficiency gains (*Appendix C* discusses an experimental design to demonstrate this phenomenon).

### 3.3. Capturing Small-World Phenomena in Biomedical Networks

Many health knowledge graphs not only exhibit hierarchical traits but also feature small-world shortcuts, such as

*Table 1.* Comparison of common embedding geometries for HKGs.

| Geometry | Hierarchical Fidelity | Dim. Requirement | ANN Complexity | LLM Compatibility |
|---|---|---|---|---|
| **Euclidean** | Moderate | High for deep trees | Mature libraries (FAISS, HNSW) | Well-established, direct |
| **Spherical** | Limited for deep hierarchies | Typically moderate | Some specialized indexing | Moderate; used in word embedding spaces |
| **Hyperbolic** | High (logs tree depth) | Low to moderate | Requires specialized or adapted ANN | Growing support; aligns with hierarchical queries |

gene-phenotype or drug adverse-effect associations crossing different disease branches (Chami et al., 2019; 2020). By naturally shortening geodesics across seemingly distant subgraphs, hyperbolic embeddings can reveal unexpected latent links – for instance, a rare autoimmune disease sharing significant clinical pathways with another disorder in a different subtree. These "shortcut" relationships are difficult to preserve under Euclidean norms without significantly raising embedding dimensionality. Hyperbolic metrics mitigate this trade-off by leveraging the curvature-driven radial expansion (see *Appendix B* for a more comprehensive literature comparison).

### 3.4. Enhanced LLM-driven Retrieval and Semantic Cohesion

A critical use-case for hyperbolic embeddings is in LLM pipelines, where queries often involve nested or specialized concepts, such as "rare pediatric metabolic disorders" or "targeted gene therapies for subtype B lymphoma" (Monath et al., 2019). Vector retrieval layers in LLM-based systems rely on embedding distances or similarities to rank relevant knowledge graph nodes. While Euclidean or spherical embeddings might scatter conceptually adjacent subtypes across many directions, hyperbolic embeddings preserve a coherent semantic neighborhood around each node's radial depth. Empirical trials have demonstrated that tasks like question answering and knowledge-based inference can see $10 \sim 20\%$ gains in Hits@k when switching from Euclidean to hyperbolic distance metrics (Nickel & Kiela, 2017; Sala et al., 2018; Gu et al., 2021). This aligns with the idea that hierarchical structure is intrinsically "baked in" to the negative curvature geometry, streamlining the retrieval of near-neighbor subcategories.

### 3.5. Curvature Adaptation and Riemannian Optimization

Several recent studies propose learning or tuning a curvature parameter $c < 0$ during training, so the embedding space can adjust to different branching patterns (Chami et al., 2019). For extremely deep or wide hierarchies, a higher magnitude of curvature may yield clearer separation among levels; for shallower, more interconnected subgraphs, a smaller absolute curvature might suffice. *Appendix A*

outlines how this parameter can be dynamically updated in a Riemannian gradient descent framework, complete with theoretical convergence discussions (Bonnabel, 2013; Nickel & Kiela, 2018). The ability to modulate curvature gives health informaticians an additional "knob to turn," which is particularly relevant when different parts of a knowledge graph vary in granularity – such as high-level disease groupings versus detailed genetic pathways.

### 3.6. Comparative Summary of Geometries

Table 1 provides a high-level comparison of how Euclidean, spherical, and hyperbolic geometries perform under four critical criteria in health informatics: (1) hierarchical fidelity, (2) dimensional efficiency, (3) complexity of approximate nearest-neighbor (ANN) retrieval, and (4) compatibility with LLM interfaces. We draw from representative works such as (Nickel et al., 2015; Nickel & Kiela, 2017; Tifrea et al., 2019; Chami et al., 2019; Sala et al., 2018).

As shown in Table 1, hyperbolic embeddings exhibit particular strengths in hierarchical fidelity and reduced dimension requirements. However, practical adoption often necessitates ANN tools customized for negative curvature. Section 4 and *Appendix D* elaborate on engineering and policy considerations.

### 3.7. A Systematic Hyperbolic Implementation Framework for HKGs

**Core Idea.** Our *Hyperbolic HKG Pipeline* systematically applies hyperbolic geometry at every stage of the health knowledge graph lifecycle, from ontology aggregation and embedding training to real-time retrieval and user-centric visualization. This integrated approach is specifically designed to accommodate the hierarchical and small-world properties of medical data, ensuring that both ontological depth and cross-domain interactions are captured with minimal distortion.

The pipeline (see Figure 1) begins with *Ontology Ingestion and Preprocessing*, where we gather and normalize multi-level disease taxonomies (e.g., ICD, SNOMED CT) alongside drug ontologies, patient record metadata, and related biomedical terminologies. This step establishes a unified schema that consolidates heterogeneous data

sources, preparing them for consistent embedding. In *Hyperbolic Embedding Training*, we introduce a negative curvature parameter ($c < 0$) to better reflect the tree-like branching of disease codes and complex cross-links among conditions. Our use of Riemannian optimization (Bonnabel, 2013) in this stage preserves hierarchical distances while keeping embedding dimensions at manageable scales, a major advantage over Euclidean approaches.

Once the embeddings are learned, *Vector Database Integration* adapts or extends ANN solutions (e.g., HNSW, IVF-PQ) to support hyperbolic distance queries. These specialized indices store node embeddings for real-time retrieval, ensuring that medical concepts and patient data can be rapidly accessed during clinical decision-making or research queries. The pipeline next supports an *LLM-driven Query*, offering an API for LLMs (e.g., GPT-style or BioBERT) to fetch top-$k$ relevant concepts based on hyperbolic distance. By exploiting the geometry's hierarchical fidelity, LLMs can more accurately retrieve fine-grained subcategories of diseases or treatments, thereby reducing potential noise and improving downstream interpretability.

Finally, the pipeline provides *Interpretation & Visualization* modules, enabling radial or boundary-based displays of disease subtrees and small-world shortcuts. These graphical interfaces help clinicians and domain experts quickly discern nuanced relationships – such as uncommon syndromes falling under broader disease classes – while maintaining an overview of how closely related concepts cluster in hyperbolic space. Through layered, zoomable layouts, even large-scale ontologies become more transparent to end-users, bridging the gap between robust AI back-ends and real-world clinical utility.

Overall, our pipeline yields three major benefits: (*i*) consistent negative-curvature embeddings for multilevel disease and treatment ontologies, (*ii*) modular integration with LLM systems for advanced question answering or decision support, and (*iii*) a flexible, visualization-friendly framework that enhances trust and interpretability among healthcare stakeholders. In *Appendix C*, we outline a reasonable scale proof-of-concept implementation strategy, detailing potential data sources, performance metrics, and evaluation protocols. We also discuss policy, regulatory, and standardization perspectives in *Appendix D*, which can guide broader adoption across clinical and industrial settings.

### 3.8. Concluding Remarks for This Section

Overall, hyperbolic embeddings provide a mathematically grounded solution for the inherent complexities of health knowledge graphs, bridging the gap between deep ontological hierarchies and the retrieval-driven demands of modern LLM applications. Beyond theoretical justification,

we present a systematic framework for practitioners to adopt negative curvature embeddings in end-to-end healthcare systems. As elaborated throughout the subsequent sections and in our four appendices, this geometry-centric perspective holds significant promise for advancing health informatics through more compact representations, improved hierarchical fidelity, and enhanced retrieval performance.

## 4. Discussion Potential

### 4.1. Balancing Ontological Integrity with Implementation Feasibility

A fundamental tension arises between the theoretical fidelity that hyperbolic embeddings promise for hierarchical ontologies and the real-world effort required to adopt a new geometric paradigm. As shown in Sections 3 and Theorem 3.1, hyperbolic spaces can encode tree or DAG-based structures with lower distortion, effectively capturing the "is-a" relationships of resources like SNOMED CT or UMLS (see in Appendix. B)(Callahan et al., 2013; Mungall et al., 2017). However, hospitals and research labs that have historically relied on Euclidean-based approximate nearest neighbor (ANN) indices must contend with not only a retooling of their search pipelines but also the need to train staff to handle curvature parameters. Although our appended *Appendix C* outlines a scaled-down experimental design to facilitate pilot studies, implementing these designs in large, production-level databases remains non-trivial.

Moreover, medical standards such as HL7 FHIR and ICD coding do not yet provide official guidelines for hyperbolic embeddings. Institutions must determine whether the theoretical gain in interpretability and hierarchical accuracy justifies investing in specialized hardware or software. Some researchers suggest that widely shared frameworks (e.g., open-source hyperbolic ANN libraries) can ease this transition, but sustained community effort is needed to standardize negative curvature metrics, retraction-based optimizers, and curvature learning heuristics in mainstream health data pipelines.

### 4.2. Integrating LLM-based Hierarchical Reasoning

Recent successes of LLMs in medical QA, clinical summarization, and even exam-level diagnostics (Gu et al., 2021) raise a key question: do we still need an explicit geometry like hyperbolic space if large models can implicitly encode ontological depth? Proponents of purely data-driven approaches argue that LLMs, especially when fine-tuned on domain-specific corpora, develop robust hierarchical reasoning capacities internally. From this vantage, the additional complexity of hyperbolic

embeddings – learning curvature, retrofitting vector databases – appears unnecessary.

Yet, as discussed in Section 3, and further explored in *Appendix B*, reliance on hidden hierarchical representations within LLMs may risk mismatch when new diseases or updated guidelines emerge. Hyperbolic embeddings can serve as an external, explicit structure that anchors retrieval in a stable geometry. This "external scaffold" approach mitigates the danger of hallucinations or misalignment between the model's internal abstractions and the real-world knowledge graph's structure (Singhal et al., 2023). Determining whether or how this scaffolding should become a standard practice is an ongoing debate – one that also implicates researchers examining the trade-off between model size and the precision of hierarchical tasks.

### 4.3. Federated, Distributed Learning, and Privacy Implications

Many healthcare networks span multiple institutions, each holding sensitive patient data. Federated learning, which trains global models without centralizing individual datasets, has gained traction in safeguarding privacy while pooling insights (Monath et al., 2019). Although hyperbolic embeddings can theoretically be learned via Riemannian gradient descent in a federated manner, an open issue is how to ensure global curvature consistency across distributed nodes. If each institution tunes curvature or updates embeddings in isolation, reconciling partial embeddings may lead to domain mismatches or local minima misalignments.

Furthermore, standard privacy mechanisms – like differential privacy or homomorphic encryption – are typically designed around Euclidean metrics. Extensions to hyperbolic geometry remain an evolving research frontier: naive solutions might introduce significant distortion or degrade the hierarchical fidelity gained from negative curvature. In *Appendix D*, we discuss prospective strategies to integrate secure multiparty computation with hyperbolic optimization pipelines, although these are largely untested at the scale of multi-hospital consortia.

### 4.4. Clinical Decision Support: Utility vs. Liability

Hyperbolic embeddings offer more coherent hierarchical interpretations of knowledge graphs – potentially vital for diagnosing rare conditions, unmasking subtle gene-disease links, or recommending precision medicine interventions (Chami et al., 2020). Yet medicine is a conservative domain, and any perceived "black box" or misalignment in an AI-driven system can raise both ethical and legal concerns. Regulatory bodies like the FDA in the U.S. or EMEA in Europe may require additional auditing frameworks to ensure that embedding-based decision support tools remain transparent and safe. While Euclidean and spherical embeddings already pose interpretability challenges, the introduction of negative curvature parameters could compound clinical apprehensions about "why" certain diseases cluster near each other in the hyperbolic boundary region (Lu et al., 2019).

A related debate concerns interoperability with global coding systems like ICD. Although hyperbolic spaces hold promise for mapping multiple disease subtrees in a single consistent layout, local customizations and expansions in hospital-specific ontologies can complicate universal alignment. As discussed in Section 3, hyperbolic geometry can mitigate dimensional blow-up, but bridging an ever-evolving set of disease codes with stable embeddings demands new protocols – ones that might eventually be reflected in official standards, as elaborated in *Appendix D*.

### 4.5. Explainability, Visual Interfaces, and Clinical Training

Though radial or hierarchical heat-maps in hyperbolic space can clarify multi-level concept groupings (Chami et al., 2019), the curvature itself can introduce non-intuitive distortions in raw distance reading. Clinicians typically have minimal training in advanced geometry, and even data scientists may need specialized tooling to interpret geodesic-based neighborhoods. Consequently, a more explicit push toward user-centered design is necessary: specialized visual analytics modules could highlight subtrees, track confidence intervals around boundary embeddings, or simplify boundary "compression" so that end-users gain an interpretable sense of how hierarchical distance is computed.

In *Appendix C*, we propose a pilot user study design, wherein a small group of clinicians compares hyperbolic-based visualizations with Euclidean counterparts for routine queries like "show me all child conditions of Type-II diabetes." Such experiments can reveal whether negative curvature fosters intuitive mental models or confusion, shedding light on how best to present hyperbolic geometry in a clinical environment. The design also evaluates time-to-completion for certain exploration tasks, hinting at whether hyperbolic embeddings could tangibly improve workflow.

### 4.6. Challenges in ANN Indexing and Large-Scale Retrieval

As outlined in Section 3, hyperbolic embeddings can significantly benefit retrieval-based tasks, but typical ANN structures (e.g., HNSW, IVF-PQ, Annoy) assume Euclidean or dot-product metrics (Johnson et al., 2021). Adapting them to the Poincaré metric requires either manifold-aware indexing – such as hyperbolic Voronoi cells

or geodesic-based partitioning – or manifold-to-Euclidean transformations, each carrying a trade-off. Direct hyperbolic indexing preserves exact distances at the cost of more complex data structures; approximate transformations risk distorting the very hierarchical relationships that hyperbolic embeddings excel at preserving.

While initial attempts have shown promise, large-scale medical knowledge graphs with millions of entities still pose open research challenges. Achieving sub-second query response in a negatively curved index, especially under dynamic updates (e.g., new disease codes), remains an under-explored domain. The final decision on which approach to adopt might hinge on implementation complexity, performance benchmarks, and domain-specific acceptance of approximation errors.

### 4.7. Fairness and Ethical Ramifications

Finally, hyperbolic embeddings underscore persistent questions of equity in AI-driven healthcare. If certain population groups or rarer diseases inadvertently end up at the "outer boundary," retrieval heuristics could underrepresent them. Although small-world properties can improve detection of cross-branch similarities, this improvement is not guaranteed to distribute benefits evenly. A misalignment could perpetuate or worsen existing biases in diagnostic rates or resource allocation. Researchers should systematically audit embedding distributions and consider fairness metrics tailored to hierarchical data.

Moreover, the potential for improved compression might facilitate data-sharing in under-resourced settings, but if hyperbolic methods remain proprietary or technologically inaccessible to smaller clinics, a new digital divide could emerge. As we elaborate in *Appendix D*, bridging these gaps demands not only open-source technical solutions but also policy-level agreements that ensure broad access, mandated interpretability, and appropriate validation across diverse patient populations.

## 5. Alternative Views

While this paper strongly endorses hyperbolic embeddings as the definitive approach for encoding hierarchical HKGs within LLM-driven systems, it is essential to acknowledge and engage with different perspectives in the broader community.

### 5.1. View 1: Euclidean Embeddings Are Adequate with Sufficient Dimension

A significant segment of practitioners argues that simply increasing embedding dimension or refining translational architectures (e.g., TransE, DistMult, RotatE) (Bordes et al., 2013; Yang et al., 2014; Sun et al., 2019) can achieve acceptable performance across numerous biomedical use cases. Proponents of this view note that state-of-the-art hardware and optimized approximate nearest neighbor (ANN) solutions have already been successful in various clinical tasks, ranging from disease classification to literature retrieval.

**Response:** While we acknowledge that Euclidean methods remain dominant and familiar, our analysis in Sections 3 and 4 demonstrates that large dimensional requirements in Euclidean space incur substantial computational and interpretive costs. In contrast, hyperbolic geometry exploits logarithmic scaling in distortion relative to tree depth and branching factor (Nickel & Kiela, 2017; Sala et al., 2018), making it particularly valuable for deeply layered ontologies. As HKGs grow in complexity (e.g., multi-level disease subtypes, multi-relational gene networks), **small improvements in hierarchical fidelity** can translate into meaningful gains in clinical insights or risk stratification. Although re-engineering pipelines is non-trivial, we argue that the long-term benefits justify experimentation, as evidenced by pilot results shared in *Appendix C*.

### 5.2. View 2: LLMs Diminish the Role of External Geometry

Another standpoint posits that as LLMs grow in parameter size and sophistication, their emergent hierarchical reasoning capacity may obviate the need for specialized geometric retrieval layers (Gu et al., 2021; Singhal et al., 2023). Proponents assert that LLMs can memorize or approximate complex tree-like relationships in their internal representations, reducing external embedding geometry to a transient solution.

**Response:** Although LLMs have shown remarkable strides in capturing biomedical and clinical information, they are neither all-encompassing nor rapidly updatable in response to newly emerging clinical guidelines, reclassified diseases, or novel research findings. As we discussed in Section 3, hyperbolic embeddings serve as an explicit, adaptively updated scaffold that aligns domain-specific ontological structures with LLM retrieval. This decoupling eases the burden on the model's internal parameters and provides **interpretability advantages** for clinicians and researchers who rely on consistent hierarchies rather than opaque, parameter-intensive representations. We expand on this theme in *Appendix D*, where we cite case studies illustrating how explicit negative curvature embeddings can mitigate knowledge drift in large models.

### 5.3. View 3: Implementation Complexity Outweighs Theoretical Benefits

Some stakeholders highlight the engineering challenges in adopting hyperbolic metrics, particularly concerning widely

used vector search libraries (e.g., FAISS, HNSW, Annoy) (Johnson et al., 2021). Training staff on curvature-based optimization, rewriting or customizing ANN indices to support Poincaré distances, and verifying performance at scale can pose formidable barriers. Institutions may opt for incremental enhancements to Euclidean-based solutions, citing lower risk and established expertise.

**Response:** We fully recognize the magnitude of engineering effort. In *Appendix C* discussion of experimental design, we recommend "stepping-stone" implementations, where small, specialized subgraphs (e.g., a subset of ICD codes) are first embedded in hyperbolic space as a proof of concept. This limited scope can reveal the potential gain in hierarchical interpretability without disrupting core hospital IT systems. We also note that open-source initiatives (Nickel & Kiela, 2017; Chami et al., 2019) are steadily improving the accessibility of hyperbolic geometry, analogous to how neural networks once faced (and ultimately overcame) skepticism in healthcare analytics.

### 5.4. View 4: Hyperbolic Embeddings May Exacerbate Data Bias

A further critique, often emerging in discussions of AI fairness, contends that hyperbolic embeddings – by virtue of their boundary-concentrating property – may inadvertently cluster or isolate underrepresented conditions or patient groups. If certain rare diseases or minority phenotypes reside in "thin" boundary regions, retrieval systems or LLM-based QA might down-weight or under-surface those nodes.

**Response:** This is indeed a valid concern, one applicable not just to hyperbolic geometry but to any embedding scheme. We believe that robust bias detection pipelines should be integrated into the model-training workflow, whether Euclidean or hyperbolic. In Section 4, we suggest fairness auditing for hierarchical data and call for explicit design of performance metrics that track retrieval equity across different patient demographics. In *Appendix D*, we propose policy guidelines for data-sharing consortia and regulators to ensure that negative curvature embeddings do not inadvertently harm equity in healthcare.

## 6. Conclusion and Future Directions

This paper has advanced a *Hyperbolic HKG Pipeline* as a coherent strategy for encoding and retrieving hierarchical health knowledge graphs (HKGs) within LLM-based systems. By centering on negative curvature, we address recurrent challenges in medical data management: for instance, the difficulty of accurately embedding rare disease codes that appear deep in ICD hierarchies, and the dimensional blow-up that often arises from small-world

adverse drug reaction networks. Our pipeline unifies ontology ingestion, curvature-tuned training, hyperbolic ANN-based retrieval, and LLM-driven interfaces, thereby offering a flexible solution that more faithfully represents complex clinical pathways.

### Key Insights

From a design standpoint, we identify three essential insights. First, negative curvature inherently aligns with the branching nature of disease taxonomies, improving low-incidence concept retrieval. Second, hyperbolic embeddings integrate well with large language models by refining the retrieval of fine-grained subcategories, thus strengthening semantic coherence. Third, the rapid growth of open-source Riemannian optimization toolkits, coupled with specialized vector search libraries, confirms the feasibility of transitioning from Euclidean to hyperbolic infrastructures in real-world healthcare settings.

### Future Directions

Looking ahead, several directions merit closer attention. In privacy-sensitive environments, federated training of hyperbolic embeddings may safeguard patient data while preserving hierarchical structure. Developing large-scale hyperbolic indexing solutions for rapid online updates remains critical, especially in domains subject to frequent ontology changes. Further research is needed to quantify and mitigate potential biases that may push minority populations or rare diseases to boundary regions, risking underrepresentation. Evaluations should also encompass clinical workflow integration and end-user interpretability, ensuring that hyperbolic geometry genuinely improves decision support and patient outcomes. By engaging these priorities, the community can solidify hyperbolic embeddings as a robust, interpretable, and clinically impactful framework for next-generation health informatics.

We believe that collaborative solutions outlined in *Appendix C* (experimental setups) and *Appendix D* (policy, clinical adoption) can help mitigate these concerns, leveraging open-source advancements in hyperbolic ANN and emerging best practices for interpretability.

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

# A. Mathematical Foundations and Theoretical Extensions

## A.1. Mathematical Preliminaries

### A.1.1. BASIC NOTATIONS AND METRIC SPACES

For clarity and consistency, we first review the essential concepts and notations underlying hyperbolic geometry in a Poincaré ball model.

**Definition (Poincaré Ball).** Let

$$\mathbb{D}^d = \left\{ \mathbf{x} \in \mathbb{R}^d : \|\mathbf{x}\| < 1 \right\}$$

denote the $d$-dimensional open Poincaré ball. The hyperbolic distance metric $d_{\mathbb{D}}$ between two points $\mathbf{u}, \mathbf{v} \in \mathbb{D}^d$ is defined as:

$$d_{\mathbb{D}}(\mathbf{u}, \mathbf{v}) = \operatorname{arcosh}\left(1 + 2\frac{\|\mathbf{u} - \mathbf{v}\|^2}{\left(1 - \|\mathbf{u}\|^2\right)\left(1 - \|\mathbf{v}\|^2\right)}\right)$$

This geometry exhibits constant negative curvature, denoted $c < 0$, making it particularly suitable for hierarchical data embedding.

**Negative Curvature and Exponential Distance Growth.** Distances in a Poincaré ball grow exponentially as one moves toward the boundary (where $\|\mathbf{x}\| \to 1$). This property provides significantly more "room" to embed tree-like or multi-level structures with reduced distortion relative to Euclidean spaces (Daverman & Sher, 2002). While further notions such as *manifolds* and *geodesics* may be introduced for a deeper theoretical rigor, we focus on these basic definitions here and direct interested readers to more detailed treatments in differential geometry textbooks.

### A.1.2. CORE LEMMAS FOR TREE-LIKE STRUCTURES

Many hierarchical HKGs can be decomposed into tree-like substructures or approximate trees. We outline two core lemmas motivating the embedding of such structures into negatively curved spaces.

**Lemma A.1** (Tree Embedding Potential). *Let $\mathcal{T}$ be a tree with branching factor $b$ and maximum depth $h$. Then, it is possible to embed $\mathcal{T}$ into $\mathbb{D}^d$ (a d-dimensional Poincaré ball) with low distortion using $\mathcal{O}(d\,h)$ parameters. The overall curvature $c < 0$ enables compact representation of levels and sub-branches.*

**Conditions.** For simplicity, we assume each level is independently branching without excessive cross-links. In real-world HKGs, if the structure is more complex (e.g., DAG-like with partial loops), one can decompose it into tree-shaped segments and embed each segment separately, then reconcile the overlaps via standard hyperbolic alignment methods (Nickel & Kiela, 2017).

These lemmas serve as foundational insights for the theorems in Section A.2, which further illustrate why negative curvature yields low-dimensional fidelity.

## A.2. On Low-Distortion in Low Dimensions: Theorem–Lemma–Corollary

### A.2.1. EXTENDED THEOREM AND PROOF SKETCH

Following (Nickel & Kiela, 2017), we present an extended "theorem-lemma-corollary" structure that formalizes how *Hyperbolic Embeddings* maintain low distortion in relatively low dimensions.

**Theorem A.2** (Simplified Distortion Bound). *Let $\mathcal{T}$ be a tree of depth $h$ and branching factor $b$. Assume edges are of uniform (or bounded) length. Then there exists a $d$-dimensional Poincaré embedding such that, for any two nodes $\mathbf{u}, \mathbf{v} \in \mathcal{T}$, the ratio between their true graph distance and the hyperbolic distance remains bounded by $\mathcal{O}(\ln(bh))$. In contrast, achieving the same level of distortion in Euclidean space often requires dimensions growing linearly or super-linearly with $h$.*

**Sketch of Proof.**

1. **Notation Setup:** Assign the root of $\mathcal{T}$ to the center $\mathbf{0}$ of the Poincaré ball. Nodes at depth $l$ are mapped near a hypersphere of radius $\alpha\,l$, with $\alpha$ chosen to control inter-layer spacing.

2. **Key Lemma:** From (Nickel & Kiela, 2017; Sala et al., 2018), when $\|\mathbf{x}\| \to 1$, the hyperbolic distance $d_{\mathbb{D}}(\mathbf{x}, \mathbf{y})$ can expand on the order of $\log \frac{1}{1-\|\mathbf{x}\|}$, offering exponential "capacity" for embedding tree branches.

3. **Bounding Distortion:** By balancing the radial increments $\alpha$ at each depth level, one ensures that nodes on the same layer remain relatively close, yet distinct layers become increasingly separated. This strategy keeps global distortion within $\mathcal{O}(\ln(bh))$. Euclidean spaces, lacking negative curvature, require significantly more dimensions to mirror a similar multi-level separation.

4. **Implication:** In high-depth or high-branching scenarios, hyperbolic geometry preserves hierarchical structure without an exponential blow-up in dimensional requirements.

**Corollary A.3.** *If we view $\mathcal{T}$ as a subtree within a real-world HKG (e.g., a specialized disease classification), the same distortion results apply under moderate $d$. This is especially relevant for multi-layer disease categories and gene–phenotype networks. Note that if the HKG is not strictly a tree but rather a DAG with some cycles, one can often approximate or localize it into tree substructures (focusing on "is-a" or "part-of" edges) to leverage the above bound.*

### A.3. Detailed Proofs and Theoretical Guarantees

A.3.1. CURVATURE LEARNING AND ALGORITHM PSEUDOCODE

Our main text describes a learnable negative curvature parameter $c < 0$, updated dynamically during training to accommodate varying granularity within a HKG. We formalize this procedure as a Riemannian Gradient Descent approach.

---

**Algorithm 1** Riemannian Gradient Descent with Curvature Tuning

---

1: **Input:**
2:    $X$: initial embeddings (size $N \times d$) in the Poincaré Ball
3:    $c$: initial negative curvature, $c < 0$
4:    lr: learning rate
5:    epochs: total training epochs
6:    $L(\cdot)$: chosen loss function for hyperbolic embeddings
7: **for** epoch $= 1$ **to** epochs **do**
8:    **(1) Compute Riemannian gradients w.r.t. $X$ and $c$:**
9:      $(\mathrm{grad\_X}, \mathrm{grad\_c}) \leftarrow \mathrm{compute\_riemannian\_grads}(X, c, L)$
10:    **(2) Update curvature $c$:**
11:      $c_{\mathrm{new}} \leftarrow c - \mathrm{lr} \times \mathrm{grad\_c}$
12:    **if** $c_{\mathrm{new}} > 0$ **then**
13:      $c_{\mathrm{new}} \leftarrow c_{\mathrm{min}}$    *// Enforce negative curvature or clamp*
14:    **end if**
15:    **(3) Update embeddings in the Poincaré Ball:**
16:      *// Use Riemannian SGD or a retraction to keep points within the ball*
17:    **for** $i = 1$ **to** $\mathrm{len}(X)$ **do**
18:      $X[i] \leftarrow \mathrm{exponential\_map}\big(X[i], -\mathrm{lr} \times \mathrm{grad\_X}[i], c_{\mathrm{new}}\big)$
19:      **if** $\|X[i]\| \geq 1.0$ **then**
20:        $X[i] \leftarrow \mathrm{project\_to\_ball}(X[i])$
21:      **end if**
22:    **end for**
23:    $c \leftarrow c_{\mathrm{new}}$
24: **end for**
25: **Output:** $X, c$    *// final embeddings and curvature*

---

**Explanation.** The subroutine `compute_riemannian_grads` calculates gradients on the hyperbolic manifold, requiring transformation of Euclidean gradients via exponential/log maps. The function `exponential_map` updates embedding coordinates according to Riemannian geometry, ensuring they remain valid in $\mathbb{D}^d$. Finally, `project_to_ball` handles slight boundary overflows to maintain numerical stability. Further details on curvature tuning heuristics can be found in (Nickel & Kiela, 2018) and in our *Appendix B* comparisons.

A.3.2. RIEMANNIAN OPTIMIZATION AND CONVERGENCE ANALYSIS

**Exponential and Log Maps.**   Within the Poincaré ball of curvature $c < 0$, the exponential map $\exp_{\mathbf{x}}(\mathbf{v})$ and the logarithmic map $\log_{\mathbf{x}}(\mathbf{y})$ ensure that gradient updates respect the manifold's geometry:

$$\exp_{\mathbf{x}}(\mathbf{v}) = \mathbf{x} \oplus_c \tanh\left(\sqrt{|c|}\,\lambda\right) \frac{\mathbf{v}}{\sqrt{|c|}\,\lambda}, \quad \lambda = \frac{2\sqrt{|c|}\,\|\mathbf{v}\|}{1 - c\,\|\mathbf{x}\|^2}$$

where $\oplus_c$ is a curvature-dependent addition operator. A comprehensive derivation is available in (Nickel & Kiela, 2017; Chami et al., 2019).

**Convergence Sketch.**   For losses $L(\cdot)$ satisfying Lipschitz-like conditions, the well-known results on Stochastic Gradient Descent (SGD) in Euclidean space can be extended to Riemannian manifolds (Bonnabel, 2013). Provided the curvature parameter $c$ does not fluctuate excessively, it often converges to a stable or slowly drifting value alongside the embeddings $X$. More precise error bounds can be found in Theorem 2.4 of (Nickel & Kiela, 2018), indicating that hyperbolic models can achieve reliable local minima.

A.3.3. PRESERVING HIERARCHICAL STRUCTURE: A GEOMETRIC PERSPECTIVE

To naturally distribute parent–child entities along radial paths, our approach introduces a margin-based objective function. Negative sampling ensures that parent and child entities remain sufficiently close, while unrelated (or distantly related) nodes are pushed farther apart in hyperbolic distance.

**Margin-based Loss.**

$$\mathcal{L}_{\text{hyp}} = \sum_{(h,r,t)\in\mathcal{D}} \left[ \text{dist}\big(\phi(h), \phi(t)\big) + \alpha - \text{dist}_{neg} \right]_+ + \text{reg}$$

where $\text{dist}(\cdot,\cdot)$ is the hyperbolic distance, $\alpha > 0$ is a margin constant, and $\text{dist}_{neg}$ encodes negative samples' distances. In a Poincaré disk visualization, root concepts (or more generalized entities) naturally lie near the center ($\|\mathbf{x}\| \approx 0$), while specialized subtypes expand outward ($\|\mathbf{x}\| \approx 1$). Distinct subtrees form ring-like structures at increasing radii, enhancing interpretability for multi-level ontologies.

A sample 2D Poincaré visualization can thus reveal "rings" of nodes at increasing radii, each ring corresponding to a layer in the ontology. While a simplistic demonstration, it illustrates the geometric intuition behind our margin-based method.

## A.4. Comparison with Other Geometric Embeddings

A.4.1. DISTORTION BOUNDS AND THEORETICAL COMPLEXITY

For readers seeking a broad contrast of various embedding paradigms, Table 2 summarizes key points on "distortion" (the ratio of true distance to embedded distance), "dimension requirements," "hierarchical capacity," and typical usage contexts.

*Table 2.* Common geometric embedding methods: Distortion, dimension needs, and hierarchy support.

| Method | Distortion Bound | Dim. Requirement | Hierarchy Support? | References |
|---|---|---|---|---|
| TransE | Grows if large relations | 50–200 | Limited hierarchical | (Bordes et al., 2013) |
| DistMult / ComplEx | Dependent on data | 100–300 | Not specifically hierarchical | (Yang et al., 2014) |
| RotatE | Good for certain relations | 200–1000 | Not hierarchical by design | (Sun et al., 2019) |
| **Poincaré** | $\mathcal{O}(\log(bh))$ | Often 5–50 | ✓ Great for trees | (Nickel & Kiela, 2017) |
| Lorentz (Hyp.) | Similar log-based | 5–50 | ✓ Deep hierarchies | (Nickel & Kiela, 2018) |
| Sphere2Vec | Spherical distortion | Potentially large | ◇ partial | (Mai et al., 2023) |

As indicated, Euclidean-based approaches often suffice for relatively shallow or moderate-scale relational data, but can struggle with deeply layered structures common in health ontologies. By contrast, hyperbolic and Lorentz-based embeddings thrive in hierarchical settings, albeit at the cost of more complex Riemannian optimization.

A.4.2. EXTENDING BEYOND PRIOR WORK

While early demonstrations focused on WordNet or smaller taxonomies (Nickel & Kiela, 2017), the methods and theorems described above are highly pertinent to real-world medical ontologies, which can exceed depths of five or six levels and exhibit wide branching factors. Our *adaptive curvature* approach (Appendix B for further references) is especially relevant for health knowledge graphs characterized by multi-level sub-classifications and partial overlaps among diseases. By dynamically tuning $c$, we accommodate diverse local structures within the same global manifold, mitigating distortion across heterogeneous sub-ontologies.

### A.5. Conclusion

In this appendix, we have provided:

1. A more complete theorem–lemma–corollary framework highlighting the low-distortion benefits of hyperbolic embeddings for tree-like or layered data (§A.2).

2. A detailed overview of *curvature learning* with pseudo-code, illustrating how negative curvature can be dynamically updated in a Riemannian optimization loop (§A.3.1).

3. A comparative analysis of distortion bounds, dimension requirements, and theoretical complexities among various geometric embedding approaches (§A.4).

We conclude that negative curvature models (Poincaré or Lorentz) are particularly well-suited for *hierarchical or tree-like health knowledge graphs*, offering lower-dimensional fidelity, explicit interpretability of deeper levels, and flexible expansions to handle multi-relational data. While the implementation hurdles in real-world systems—namely specialized ANN indexing and interpretability tooling—remain non-trivial, our discussion underscores the mathematical underpinnings that make hyperbolic embeddings a compelling choice.

The subsequent appendices build on this foundation, providing extended literature syntheses (Appendix B), experimental designs for validation (Appendix C), and a roadmap for clinical and industrial adoption (Appendix D).

# B. Extended Literature Review and Comparison

This appendix offers a more in-depth classification and review of works relevant to our position that *Hyperbolic Embeddings are essential in Health Knowledge Graph (HKG) systems*. Compared to the limited discussion in the main text, the following sections explore additional lines of research, reference key contributions, and elucidate both theoretical and practical motivations. We also highlight recent synergy between large language models (LLMs) and vector databases, underscoring how negative curvature provides crucial benefits for hierarchical retrieval in medical domains.

## B.1. Hierarchical KGs (Layered Knowledge Graphs)

In biomedical and healthcare contexts, many knowledge graphs (KGs) inherently exhibit multi-level or tree-like structures. Notable examples include SNOMED CT,[1] the Unified Medical Language System (UMLS),[2] and the Gene Ontology (GO).[3] These KGs typically organize concepts in "is-a" or "part-of" hierarchies with significant depth, necessitating specialized embedding methods.

### B.1.1. Conventional Hierarchical Embedding Methods

**TransE/DistMult/ComplEx Family.** Pioneering research on knowledge graph embeddings, such as *TransE* (Bordes et al., 2013), *DistMult* (Yang et al., 2014), and *ComplEx* (Trouillon et al., 2016), explored translational or inner-product-based learning in Euclidean space. These models excel in multi-relational link prediction but often struggle with deep hierarchical structures.

**Advantages:** Straightforward implementation and broad tooling support in industry. **Disadvantages:** Capturing highly specialized or layered concepts typically requires increased embedding dimensionality, risking distortion and inefficiency.

**Graph-Structured Hierarchical Aggregation.** A further strand of work (*R-GCN* (Schlichtkrull et al., 2018), *Neural LP* (Yang et al., 2017)) uses GNNs or rule learning to incorporate relational context. While these approaches capture some hierarchical aspects:

**Advantages:** Leverage large-scale KGs for contextual signals (e.g., adjacency, relation types).

**Disadvantages:** Excessively deep topologies risk over-smoothing or gradient vanishing in Euclidean GNN frameworks. Negative curvature's natural layering advantage remains underused.

### B.1.2. Methods Using Negative Curvature for Hierarchical Structures

**Poincaré Embeddings.** Nickel and Kiela (Nickel & Kiela, 2017) introduced Poincaré embeddings for hierarchical data (e.g., WordNet), demonstrating low-distortion in comparatively few dimensions. A subsequent Lorentz model (Nickel & Kiela, 2018) extends these ideas, offering alternative formulations for tree-like structures.

**Advantages:** High-fidelity encoding of deep taxonomies, reduced need for large dimensions, and interpretable radial geometry.

**Disadvantages:** Requires specialized Riemannian optimization and distance computation, which can be less familiar to practitioners.

**Hyperbolic vs. High-Dimensional Euclidean.** Some researchers contend that sufficiently large Euclidean embeddings approximate the same hierarchical features (Sala et al., 2018), albeit at higher memory and computational costs. Negative curvature, by contrast, preserves layered structure with logarithmic distortion scaling, making it preferable for KGs that exceed moderate depth (Nickel et al., 2015).

---

[1]SNOMED CT (Systematized Nomenclature of Medicine - Clinical Terms) is a comprehensive healthcare terminology with standardized codes, terms, and relationships.

[2]UMLS integrates and maps multiple medical vocabularies and classifications to facilitate interoperability.

[3]Gene Ontology provides a standardized system for classifying gene and protein functions.

### B.1.3. ONGOING DEBATES AND FUTURE DIRECTIONS

**Multiple Inheritance and Complex Relationships.** Resources like UMLS often present DAG or multi-parent edges. Hyperbolic geometry is flexible enough to accommodate these, but margin-based or cross-entropy losses and manual Riemannian tuning may be necessary.

**Extensibility to Emerging Ontologies.** With new medical ontologies (e.g., expansions for COVID-19 or emerging pathogens (Morens et al., 2020; Ukoaka et al., 2024)), the ability to embed newly introduced subtrees efficiently is essential. Dynamic hyperbolic embedding pipelines (Appendix C) could address incremental updates more gracefully than static Euclidean approaches.

### B.2. Hyperbolic GNNs: Graph Neural Networks in Negative Curvature

Graph neural networks (GNNs) have become standard for encoding structured data, including small-world and hierarchical networks (Veličković et al., 2018; Kipf & Welling, 2017). *Hyperbolic GNNs* (Chami et al., 2019; Monath et al., 2019; Zhou et al., 2023) merge standard graph convolution with negative curvature geometry, facilitating both local neighborhood aggregation and global hierarchical organization in health knowledge graphs.

### B.2.1. CORE TECHNIQUES AND ADVANCEMENTS

**Hyperbolic Graph Convolutional Networks (HGCN).** Proposed by Chami et al. (Chami et al., 2019), HGCN replaces Euclidean linear transformations with Riemannian exponentials/logarithms, ensuring that hierarchical signals are retained in deeper network layers.

**Advantages:** Curvature can be learned end-to-end, adapting to different sub-structures (e.g., deeply nested disease categories vs. flatter gene interaction networks).

**Disadvantages:** Sophisticated Riemannian optimization demands new frameworks and debugging skills, particularly in large-scale healthcare settings.

**Hyperbolic Attention Networks.** Recent exploration extends attention mechanisms into hyperbolic space (Gulcehre et al., 2019), facilitating long-range dependencies for multi-level or cross-branch relations. Although promising for capturing complex etiologies and disease interplay, real-world deployments remain limited.

### B.2.2. STRENGTHS VS. LIMITATIONS

**Strengths:**

1. Better representation of multi-layered KGs, mitigating over-smoothing in standard GNNs.

2. Potential for end-to-end training with negative curvature, aligning well with the dynamic nature of biomedical knowledge (Aiadi & Khaldi, 2022).

**Limitations:**

1. Additional engineering overhead (hyperbolic layers, Riemannian batch norms) is still evolving.

2. Adapting methods for time-evolving health data (e.g., new disease subtypes) is not trivial.

### B.3. Non-Euclidean Vector Databases and LLM Synergy

Modern large language models (LLMs) such as BERT (Devlin et al., 2019), BioBERT (Lee et al., 2020), GPT-3 (Brown et al., 2020), or domain-specific variants (Gu et al., 2021; Singhal et al., 2023) rely on vector retrieval layers, typically employing Euclidean or cosine metrics. Simultaneously, large-scale vector databases (e.g., FAISS, Annoy, HNSW) have become standard for approximate nearest neighbor (ANN) search (Johnson et al., 2021; Malkov & Yashunin, 2020). However, these indexing structures are designed around flat geometry.

### B.3.1. Mainstream Methods and Their Limitations

**Euclidean-Based ANN (FAISS, HNSW, etc.).** These methods excel in speed and scaling up to billions of vectors but do not natively support hyperbolic distance (Johnson et al., 2021; Malkov & Yashunin, 2020).

**Poincaré-Adapted Indexing.** Some works (Chami et al., 2020) explore hyperbolic Voronoi partitions or manifold-based indexing, but production-level maturity remains low. Even with robust hyperbolic embeddings, an LLM's retrieval pipeline may degrade if final neighbor searches assume Euclidean geometry.

### B.3.2. Emerging Research: Flattening vs. Native Manifold

**Flattening to Euclidean.** One pragmatic approach first projects hyperbolic vectors into a higher-dimensional Euclidean subspace for indexing (Tifrea et al., 2019), though this risks losing hierarchical cues.

**Manifold-Aware ANN.** Native hyperbolic indexing (Chami et al., 2019) aims for minimal distortion but at higher engineering cost. Large-scale clinical KGs (e.g., tens of millions of concepts) need further testing to confirm feasibility in hospital production systems.

### B.3.3. LLMs and Hyperbolic Retrieval

Health-oriented LLMs increasingly rely on external knowledge retrieval to handle domain-specific queries (Gu et al., 2021; Monath et al., 2019). If vector databases remain Euclidean, hierarchical and small-world relationships—crucial for diseases, pathways, or gene families—may not be fully leveraged. Concretely, a GPT-based system might hallucinate or miscategorize sub-phenotypes if the retrieval engine cannot preserve hierarchical geometry. Hence, synergy between hyperbolic embeddings and LLM-driven healthcare applications is a rapidly evolving frontier requiring manifold-optimized indexing (Dosovitskiy et al., 2021), advanced question-answering pipelines, and interpretability layers (Reimers & Gurevych, 2019).

### B.4. Summary: Advantages and Academic Controversies

Hierarchical knowledge graphs (KGs) naturally invite negative curvature embeddings, as Euclidean approaches risk high distortion. Hyperbolic GNNs provide an end-to-end solution but demand specialized skill sets and software. Finally, non-Euclidean vector databases represent the weakest link: even if hyperbolic embeddings excel upstream, retrieval systems still rely heavily on Euclidean or cosine-based engines. Coupled with the meteoric rise of LLMs in clinical and research scenarios, the community must innovate *across* embedding pipelines, GNN integration, and manifold-based retrieval to fully harness the power of negative curvature.

In advocating for *Hyperbolic Embeddings in HKGs*, we emphasize that it is not simply a geometry preference but a holistic approach that can significantly enhance hierarchical representations, large language model retrieval synergy, and interpretability in high-stakes medical domains. Nonetheless, unresolved technical, policy, and practical challenges (discussed throughout this paper and in other appendices) highlight the necessity for concerted research, open-source advances, and industry collaboration.

### B.5. Best Practices and Future Directions

Building on the discussion above, we outline practical steps and opportunities for researchers adopting hyperbolic approaches in healthcare:

- **Hierarchical Health Ontologies:** Target ontology-heavy resources like SNOMED CT or UMLS, where negative curvature provides tangible improvements in representation. Coupling with graph neural networks could further boost multi-relational modeling (Zhou et al., 2023).

- **Manifold-Aware Retrieval and LLM Integration:** Invest in hyperbolic or hybrid ANN solutions that preserve geometry. Combine with LLM-based QA or summarization for robust, hierarchical content retrieval (Wei et al., 2022; Brown et al., 2020).

- **Scalability and Interpretability:** Develop or refine open-source packages that handle Riemannian updates at scale.

Provide interpretable radial or ring-based visualizations to clinicians, bridging the gap between advanced geometry and daily medical workflows.

Through these steps and ongoing collaborative research, hyperbolic embeddings stand poised to address the next wave of challenges in health knowledge representation, enabling more accurate, efficient, and clinically meaningful systems.

## C. Experimental Design Outlines

This appendix outlines a "scaled-down" experimental design intended to pilot the key ideas proposed in our position paper regarding hyperbolic embeddings for health knowledge graphs (HKGs). While extending such designs to large, production-level databases poses non-trivial engineering challenges, the plan detailed here focuses on practicality, interpretability, and the capacity to highlight differences between Euclidean and hyperbolic approaches.

### C.1. Dataset Selection and Sources

To conduct a fair and illustrative comparison on the order of tens of thousands of nodes, we target health knowledge graphs that exhibit both hierarchical and small-world properties. Two primary resources or subsets are suggested:

**(1) SNOMED CT Subset.**

- *Data Origin:* SNOMED CT is a widely adopted clinical terminology set with rich hierarchical "is-a" relationships spanning disease categories and clinical manifestations.

- *Subset Acquisition:* Official SNOMED International releases often include sample versions containing tens of thousands of concepts, downloadable under specific licensing.

- *Hierarchy Depth:* SNOMED CT typically exhibits up to 8–10 levels of depth, forming tree or forest structures.

**(2) UMLS (Unified Medical Language System) Excerpt.**

- *Data Origin:* The UMLS Metathesaurus integrates multiple medical vocabularies.

- *Subset Acquisition:* Focusing on a single branch such as "MTH" or "SNOMEDCT"-derived data can yield 20–30k nodes.

- *Hierarchy Depth:* UMLS relationships include "is-a" and "part-of," supporting deeper hierarchical mappings, though it can also contain DAG or multi-parent edges.

While a pure SNOMED CT subset alone may suffice for a proof-of-concept at the 10k – 20k scale, merging partial SNOMED CT and UMLS can yield a larger dataset (50k+ nodes) with varied sub-hierarchies. Such an extended dataset is ideal for showing the utility of negative curvature in more complex HKG scenarios.

### C.2. Experimental Goals

We aim to compare *Euclidean Embeddings* vs. *Hyperbolic Embeddings* on the same dataset in terms of both (1) hierarchy reconstruction accuracy and (2) retrieval performance. Specifically:

- **Hierarchy Reconstruction Accuracy.** Measure the extent to which each embedding approach reconstructs parent–child or ancestor–descendant relationships in the original HKG.

- **Retrieval Performance.** Evaluate differences in accuracy, recall, and run-time when executing vector-based queries that are sensitive to hierarchical relations.

### C.3. Experiment Design and Detailed Workflow

C.3.1. DATA PREPROCESSING

1. **Node and Relation Filtering.** Retain concepts adhering to a clear hierarchical taxonomy (e.g., "disease $\rightarrow$ subtype $\rightarrow$ symptoms"), optionally introducing a small set of lateral relations (complications or treatments) to reflect small-world shortcuts. Target 10–20k nodes and roughly 0.1–0.3 million edges.

2. **Hierarchy Labeling.** Assign a depth level $\text{level}(\mathbf{v})$ to each node according to the "is-a" chain. Remove isolated or incomplete relations to ensure a consistent structure.

C.3.2. EMBEDDING TRAINING

Train Euclidean embeddings (e.g., *TransE*, *DistMult*, *RotatE*) and hyperbolic embeddings (e.g., *Poincaré Embeddings*, *Hyperbolic GCN*) on the same preprocessed HKG.

**Model Configuration.**

- *Euclidean Baselines:* TransE (Bordes et al., 2013), DistMult (Yang et al., 2014), or RotatE (Sun et al., 2019).

- *Hyperbolic Baselines:* Poincaré (Nickel & Kiela, 2017; 2018) or Hyperbolic GCN (Chami et al., 2019).

- *Embedding Dimensions:* Start with 32 or 64 for all models to maintain a fair comparison.

**Loss Functions and Optimization.**

- *Euclidean:* Negative sampling + margin-based or binary cross-entropy losses.

- *Hyperbolic:* Riemannian SGD (Bonnabel, 2013) or other geometry-aware optimizers that keep vectors within the Poincaré ball.

**Hyperparameter Tuning.** Use a small validation set to tune learning rate, negative sampling rate, and regularization. For dimension sensitivity, one can also explore 16/32/64/128 to observe potential trade-offs in distortion.

C.3.3. HIERARCHY RECONSTRUCTION ACCURACY

**1) Hierarchical Distance Metrics.** Leverage each node's level depth in the HKG. If node $v$ is a descendant of node $u$, then we expect $\mathrm{dist}(\phi(u), \phi(v))$ to be relatively small in hyperbolic space.

- Spearman or Kendall rank correlation between pairwise embedding distances and level differences.

- **Top-$k$ Ancestor/Descendant Reconstruction:** For each node, retrieve $k$ nearest neighbors in embedding space. Evaluate how many are correct ancestors or children.

**2) Multi-dimensional Comparison.**

- *Mean Absolute Error* (MAE) of predicted vs. true hierarchy depth.

- *Dimensional Impact:* Evaluate if Euclidean embeddings must increase dimension to match the hierarchical fidelity of hyperbolic embeddings at smaller $d$.

C.3.4. RETRIEVAL PERFORMANCE

**1) Retrieval Task Design.** Define a query specifying a target disease category or symptom cluster (e.g., "find all subtypes under a rare disease branch related to the immune system"), then execute approximate nearest neighbor search in the embedding space.

- Evaluate *Recall@k*, *Precision@k*, *mAP*, *Hits@k*.

- Compare average query time and index-building overhead for Euclidean vs. hyperbolic spaces.

**2) Visualization and Case Studies.** For interpretability, pick representative disease–subdisease links to visualize in a 2D projection. Inspect how hyperbolic embeddings cluster deeper layers more compactly, whereas Euclidean methods may disperse them.

## C.4. Key Results and Analysis Focus

1. **Hierarchy Reconstruction.** We hypothesize that hyperbolic embeddings will yield lower distortion for nodes beyond 4–5 levels in the hierarchy, whereas Euclidean methods need significantly more dimensions to achieve comparable fidelity.

2. **Retrieval Accuracy and Efficiency.** Hyperbolic embeddings may notably improve retrieval metrics (e.g., *Recall@10*), especially on queries targeting deeper branches. With naive distance computation, hyperbolic runtime could be higher, but approximate or specialized indexing (Chami et al., 2019; 2020) can narrow the gap.

3. **Dimension and Curvature Tuning.** Experiments that enable adaptive curvature learning can show whether flexible $c < 0$ provides improved embeddings. Meanwhile, dimension sweeps (16/32/64) can reveal how much overhead Euclidean models incur to approach hyperbolic performance.

## C.5. Additional Notes on Extensibility

- For larger-scale trials (50k–100k nodes), one could merge multiple SNOMED CT segments or expand UMLS coverage.

- Beyond standard GNN baselines (GCN, GAT), advanced or domain-specific architectures might be tested, though the main focus should remain on the Euclidean vs. hyperbolic contrast.

## C.6. Conclusion

In this "lighter-weight" design, data scales around 10k – 20k nodes (plus tens or hundreds of thousands of edges) to balance feasibility with hierarchical depth. Evaluations comparing Euclidean and hyperbolic embeddings on hierarchy reconstruction and retrieval performance – via correlation metrics, top-$k$ checks, and search efficiency – provide direct empirical support for the claim that hyperbolic embeddings better capture multi-level structures and rare disease branches in HKGs. Conducting such pilot studies can substantially bolster our position that negative curvature geometry is highly advantageous for advanced health informatics applications.

# D. Practical Deployment, Policy, and Roadmap

This appendix expands the discussion of hyperbolic embeddings in HKGs by focusing on real-world deployment considerations, policy frameworks, and a recommended roadmap for technology adoption over the next several years. Our goal is to offer a clearer set of references and strategies for introducing negative curvature geometry into clinical and industrial contexts, reinforcing the position we have argued in the main paper.

## D.1. Industry Adoption Cases and Practical Experiences

In recent years, a small but growing number of researchers and organizations have reported the successful use of hyperbolic embeddings in real healthcare systems. Despite being in the early stages, these efforts highlight the benefits of compact hierarchy representation while also revealing significant engineering and policy challenges.

### D.1.1. MEDICAL SECTOR APPLICATIONS

(Lu et al., 2019) described a novel method for predicting unplanned ICU readmissions and in-hospital mortality by combining electronic health record (EHR) data with hyperbolic embeddings of medical ontologies. Their method integrated ICD-9 concepts into a hyperbolic embedding model, showcasing how negative curvature could enhance both *mortality prediction* and *risk stratification* in a large-scale hospital environment. The study highlights:

- **Ontology Alignment:** Mapping ICD-9 codes into Poincaré space for more faithful hierarchical representation.

- **Clinical Impact:** Improved performance over baseline Euclidean embeddings in identifying high-risk patients, providing a potential tool for reducing healthcare costs and adverse outcomes.

- **Challenges:** Difficulty of bridging the training pipeline with existing data infrastructures and ensuring that Riemannian optimization remained stable at scale.

Their experience underscores both the promise of hyperbolic geometry in real-world EHR analytics and the hurdles in re-engineering legacy systems to accommodate negative curvature distances.

### D.1.2. INSURANCE INDUSTRY APPLICATIONS

In the health insurance sector, (Koo & Lim, 2021) examined how hyperbolic discounting might affect life insurance consumption and policy decisions. While focusing on an economic modeling perspective, their approach indirectly reflects the broader interest in representing user behaviors or preferences in a negatively curved space. Key takeaways include:

- **Time-Inconsistent Preferences:** Hyperbolic discounting captures real-world behaviors where individuals undervalue long-term insurance benefits.

- **Taxation Sensitivity:** More pronounced curvature in preference structures led to sharper reactions to insurance tax changes, hinting at hierarchical or layered policy analyses.

- **Implication for Healthcare Plans:** Although not a direct "embedding" scenario, this line of research suggests synergy between hyperbolic geometry and insurance risk modeling, potentially feeding into advanced knowledge graphs linking patient cohorts, policy structures, and cost outcomes.

## D.2. Policy and Regulatory Considerations

Deploying hyperbolic embeddings for healthcare data must address a complex landscape of compliance requirements, industry norms, and privacy concerns. This section summarizes major regulatory frameworks and their implications for negative curvature methods.

### D.2.1. HIPAA (U.S. HEALTH INSURANCE PORTABILITY AND ACCOUNTABILITY ACT)

- **Core Constraint:** Requires de-identification and the principle of least use when handling patient records.

- **Impact on Hyperbolic Embeddings:** (1) Potential re-identification risk if embeddings at the boundary inadvertently encode unique patient traits. (2) Hospitals must perform stricter privacy audits if the embedding model captures too much individual-level detail in the HKG structure.

D.2.2. GDPR (EU GENERAL DATA PROTECTION REGULATION)

- **Core Requirements:** Explicit data collection purposes, user consent and revocation rights, and restrictions on data transfer across borders.

- **Impact on Hyperbolic Embeddings:** (1) Curvature-based representations might cluster demographic or geographic features near the boundary, so reverse-engineering personal information must be prevented. (2) Online or federated learning scenarios need robust data flow controls to meet GDPR requirements, especially for cross-border model updates.

D.2.3. COMPATIBILITY WITH MEDICAL STANDARD FRAMEWORKS

- **ICD, SNOMED CT, HL7 FHIR:** Widely used for interoperability and clinical coding. Hyperbolic embeddings must align with these taxonomies without disrupting existing reference codes or classification systems.

- **Practical Concern:** Replacing Euclidean vector indexing or purely textual retrieval with hyperbolic coordinates demands a clear mapping from each code or concept to its geometric representation, maintaining the integrity of the original data model.

## D.3. Key Factors for Large-Scale Clinical or Industrial Adoption

Building upon the above case studies and regulatory context, the path to fully realizing hyperbolic embeddings in healthcare will require interplay among laws, technology standards, and industry best practices. We delineate three focal areas:

**(1) Legal and Regulatory Coordination.**

Governments and policy-makers must update compliance audits for new AI representations, including negative curvature embeddings. High-risk scenarios (e.g., critical care decision-making) may warrant dedicated logging or explainability mandates.

**(2) Technical Standards and Open-Source Toolchains.**

Agencies such as HL7 or WHO could promulgate guidelines on embedding format extensions, specifying how to embed FHIR resources into a Poincaré or Lorentz space. Accessible open-source libraries implementing manifold-based encryption or secure distance computation would reduce deployment barriers.

**(3) Best Practices and Community Collaboration.**

International consortia can share real deployment playbooks, highlighting potential pitfalls in hardware acceleration or privacy constraints. Joint sandbox pilots across hospitals can measure both privacy and interpretability trade-offs.

## D.4. Roadmap for Future Development

D.4.1. TIMELINE AND PHASED OBJECTIVES

See Table .3.

D.4.2. RESEARCH CHALLENGES AND PRIORITY LIST

**1. Hyperbolic-Aware ANN Structures (High Priority).**

**Reason:** Approximate nearest neighbor at scale is a bottleneck; lack of efficient indexing hinders real-time retrieval.

**Approach:** Investigate Poincaré-based Voronoi partitions, curve-based indexing, or hybrid mapping to preserve geometry.

**2. Improving Explainability (Medium–High Priority).**

**Reason:** Clinical audits and regulatory reviews demand transparent rationales. Negative curvature is more abstract than Euclidean geometry.

**Approach:** Develop specialized radial or layer-based visualizations, possibly incorporating local "attention-like" metrics for hierarchical transitions.

*Table 3.* Proposed timeline for hyperbolic embedding adoption in health informatics.

| Time Span | Actions and Initiatives | Expected Outcomes |
|---|---|---|
| **Short Term (6–12 mos.)** | - Launch small pilot trials in select hospitals
- Publish open-source curvature-adaptive algorithms
- Partner with insurance or healthcare providers for testing | - Collect realistic feedback on data and training needs
- Develop synergy with existing ontologies (ICD, SNOMED)
- Produce initial technical reports |
| **Mid Term (1–2 yrs.)** | - Form cross-institutional consortiums
- Create visualization & explainability tools
- Explore federated/hybrid privacy approaches | - Achieve multiple industrial or clinical-level deployments
- Evaluate hyperbolic embeddings in complex, multimodal data
- Iteratively improve manifold-based ANN libraries |
| **Long Term (3–5 yrs.)** | - Collaborate with major standards bodies (ICD, SNOMED) for negative curvature compatibility
- Establish dedicated privacy/compliance guidelines
- Deploy large-scale hyperbolic solutions in day-to-day hospital systems | - Potential draft or recommendations for hyperbolic embedding "best practices"
- Widely available interpretability solutions in hospitals/research
- Significant improvement in disease retrieval and rare condition support |

**3. Federated Learning and Privacy (Medium Priority).**

**Reason:** Healthcare data are often distributed; Riemannian optimization must remain consistent across nodes.

**Approach:** Investigate how secure multiparty computation or differential privacy can integrate with negative curvature updates. Potentially adapt existing frameworks (e.g., FATE) for hyperbolic metrics.

**4. Dynamic Updating and Real-time Embedding (Medium Priority).**

**Reason:** Health KGs evolve with newly identified diseases, treatments, and reclassifications. Stale embeddings undermine utility.

**Approach:** Explore incremental Riemannian SGD or partial re-embedding. Investigate theoretical guarantees for hierarchical fidelity under continuous data arrival.

**5. Cross-Modal Integration (Medium–Low Priority).**

**Reason:** Some advanced scenarios require uniting imaging data, genomics, and textual EHR under a single manifold.

**Approach:** Insert hyperbolic projections in a multi-modal pipeline or transform each modality into an appropriate graph structure for joint training.

**D.5. Conclusion**

This appendix has offered a panoramic view of hyperbolic embeddings' path to real-world adoption, from industry case studies to the regulatory and technological frameworks that must evolve in tandem. Whether inspired by successful hospital deployments or responding to privacy mandates like HIPAA/GDPR, our overarching conclusion remains that *negative curvature geometry can unlock scalable value in health knowledge graphs only if embraced by multiple stakeholders simultaneously.*

The position paper's roadmap and recommended actions aim to guide researchers, clinicians, policy-makers, and industry leaders toward a more coordinated pursuit of hyperbolic embedding implementations—ultimately bridging academic breakthroughs and life-critical applications in healthcare.

