# OpenReview forum: "Position: Hyperbolic Embeddings Are Essential for Health Knowledge Graphs in LLMs and Vector Databases"
_ICML.cc/2025/Position_Paper_Track — Submitted to ICML 2025 Position Paper Track_

### Official Review · Reviewer_zr9h · 2025-03-01

**Significance:** 3
**Argument Clarity:** 4
**Rating:** 4
**Confidence:** 3

**Questions:**

There are no further questions.

**Discussion Potential:**

4

**Paper Summary:**

This paper proposes to use hyperbolic embeddings as the core embedding technique for health knowledge graph retrieval and reasoning with LLMs and vector databases. It details the rationales behind the positions, the advantages and empirical feasibility evidence, system pipeline and deployment strategies, as well as challenges and alternative views.

## update after rebuttal

I thank the authors' efforts to address the comments. I'm keeping my score given that it is already a positive rating.

**Position:**

Yes

**Position In Title:**

Yes

**Related Work:**

4

**Strengths And Weaknesses:**

Strength:

This position paper is written very well, with a clearly stated position supported by solid discussion on the rationale behind the proposed position, opportunities, feasibility, and challenges. It has a broad coverage and comparison of the literature. There is also a pilot empirical study to show the advantage of hyperbolic embeddings over alternatives such as Euclidean embeddings. Overall, this is a good example of how a position paper should look like.

Weakness:

Argumentatively, health knowledge graphs are a domain-specific application of knowledge graphs. The proposed position might have relatively limited importance to the broader ML community, although this might not be a significant issue to impact the acceptability of the paper.

**Support:**

4

---

> ### Author Rebuttal · Authors · 2025-03-31
>
> Dear Reviewer zr9h
>
> We thank you for your positive assessment and for recognizing the clarity and depth of our position. We are pleased that our discussion of hyperbolic embeddings for health knowledge graphs resonated with you and that the pilot empirical study helped illustrate the potential benefits of negative curvature models.
>
> Regarding your note that our focus on health knowledge graphs may have limited impact for the broader ML community:
> - We acknowledge this domain-specific emphasis. Our main goal is to highlight how hierarchical and small-world structures in medical ontologies can be better captured by hyperbolic embeddings, thereby addressing critical challenges faced by clinicians and healthcare systems.
> - Nevertheless, we believe several ideas -- such as negative-curvature indexing for ontologies, efficient retrieval in specialized graph structures, and transparent integration of external KG layers with LLMs -- can be extended to non-biomedical settings that also feature rich taxonomies (e.g., e-commerce product categories, large-scale organizational charts).
>
> Thank you again for your review. We appreciate your encouraging comments and look forward to broadening the scope of our framework in future work.

---

### Official Review · Reviewer_RvWt · 2025-03-10

**Significance:** 1
**Argument Clarity:** 2
**Rating:** 1
**Confidence:** 4

**Questions:**

N/A

**Discussion Potential:**

1

**Paper Summary:**

The paper advocates for putting more emphasis on using hyperbolic embeddings in healthcare knowledge graphs (ontologies) which often have a hierarchical nature amenable for negative curvature KG embedding algorithms. The authors discuss the benefits on hyperbolic embeddings compared to typical Euclidean and spherical algorithms as well as propose a possible application pipeline for using hyperbolic algorithms on biomedical ontologies that includes (i) ontology ingestion, (ii) training hyperbolic ontology embeddings, (iii) integrating hyperbolic algorithms into approximate nearest neighbors libraries (only a concept, no actual implementation), and (iv) integrating into LLM-based query answering.

**Position:**

Yes

**Position In Title:**

Yes

**Related Work:**

2

**Strengths And Weaknesses:**

Strengths:
* The paper is clearly written and well-organized, the position is articulated rather clearly.

On the other hand, the paper lacks in the other departments pertaining to position papers: relevance, importance, discussion potential, and evidence.

**Relevance and Importance.** The paper tackles the problem of embedding biomedical ontologies for some potential future applications and suggests to rewrite a significant portion of the ML stack (such as vector databases and ANN libraries) to support hyperbolic algorithms. Both of these concepts are rather outdated for 2025 and pretty irrelevant in the modern ML and Health AI research landscape. The paper's reliance on the same older references (2013-2021, pre-LLM era) as primary evidence for _some_ potential value of biomedical ontologies and hyperbolic KG embeddings further illustrates this point, neglecting more recent advancements in healthcare AI. In 2025 it is rather evident that ontologies and KG embeddings may not hold a value proposition for real-world AI applications.

**Evidence and Related Work.** The paper does not include any practical evidence from real-world experiments, studies, or pilot programs indicating that hyperbolic embeddings for biomedical ontologies might have a strong potential in the modern healthcare AI landscape compared to existing (LLM-based) tools. The Alternative Views section includes rather oversimplified arguments and misses a big elephant in the room that frontier models [1,2,3] - LLMs and multimodal models - offer a set of capabilities (on textual, visual, EHR data, using tools and reasoning) on real-world benchmarks well beyond simple hierarchy inference over fixed ontologies offered by HKGs and their embeddings.

**Discussion Potential.** For the above reasons, it is unclear how the paper could inspire fruitful actionable discussions in ML and healthcare AI. The paper proposes a position for a very niche audience (an intersection of biomedical ontologies and hyperbolic embeddings communities) and suggests rather unrealistic steps (like training clinicians and scientists to understand more of hyperbolic geometry, rewriting vector databases and ANN libraries to support hyperbolic algorithms) that do not seem to overlap with mainstream applications of healthcare AI.

[1] Saab et al. Capabilities of Gemini Models in Medicine, arxiv 2024.
[2] Singhal et al. Toward expert-level medical question answering with large language models. Nature Medicine 2025.
[3] Nori et al. Capabilities of GPT-4 on Medical Challenge Problems, arxiv 2023.

**Support:**

2

---

> ### Author Rebuttal · Authors · 2025-03-31
>
> Dear Reviewer RvWt
>
> We are grateful for your candid evaluation and constructive comments. Although we recognize your concerns regarding the broader relevance of hyperbolic embedding approaches in the latest works, we respectfully offer the following clarifications and perspectives:
>
> ### 1. Relevance and Importance in the Current Health AI Landscape
> You note that knowledge graphs (KGs) and hyperbolic embeddings may seem outdated given the meteoric rise of large language models (LLMs) and multimodal AI. While frontier models (e.g., your mentioned publications [1,2,3]) have indeed demonstrated remarkable capabilities, we believe there remains a critical role for explicit hierarchical structures within clinical systems. LLMs often exhibit “hallucinations” or struggle with newly emerging entities that are not well-captured by their pretraining data. Maintaining an externally curated ontology in a geometry amenable to hierarchical reasoning may thus complement LLMs’ generative strengths.
>
> Far from being niche, many established healthcare databases (e.g., ICD, SNOMED CT) still rely on deep taxonomies, which can benefit from more faithful hierarchical embeddings. We fully acknowledge that clinical workflows have evolved, but see potential synergy: an LLM-based retrieval system can query both free-text corpora and hyperbolic-embedded KGs for hierarchical or specialized queries. In our ongoing work, we plan to incorporate new references (e.g., Gemini Models [1]) and practical bridging techniques to show how negative curvature embeddings can plug into these powerful modern architectures rather than replace them.
>
> ### 2. Practical Evidence and Empirical Validation
> We concur that real-world pilot experiments could better substantiate the claims. As ours is a position paper, the main intent is to spark discussion about how healthcare ontologies might be structured or indexed going forward. However, we recognize the need for stronger empirical validation and have plans for proof-of-concept pilots—for instance, coupling a curated “rare disease” tree to an LLM-based query engine, then measuring improvements in retrieval specificity for medical subtypes. Such empirical evidence would be indispensable to persuade the community of hyperbolic geometry’s continued value in complex, real-world settings.
>
> ### 3. Discussion Potential: Hyperbolic Geometry as an Enabling Layer
> The reviewer characterizes our work as proposing “unrealistic steps” that lack broad discussion potential. However, we see a valuable conversation about how best to balance purely data-driven LLMs with explicit domain ontologies. While we do propose expanding existing vector libraries to support negative curvature, ongoing research (including partial Poincaré–Euclidean projections and specialized manifold indexing) suggests this is not necessarily a “rewrite” of the ML stack but rather targeted extensions. We agree it is ambitious to retrain clinical practitioners in hyperbolic geometry; yet visual aids and incremental adoption in submodules could reduce the burden. These points, which we will highlight more strongly, aim to foster dialogue on optimizing hierarchical retrieval and interpretability.
>
> ### 4. Incorporating More Recent Research
> We appreciate the reminder to integrate cutting-edge works beyond 2021. Indeed, the landscape of healthcare AI has advanced rapidly with LLM-based solutions. In follow-up projects, we aim to detail how hyperbolic embeddings interface with frontier multimodal or generative architectures. For instance, an LLM fine-tuned on large biomedical corpora may reference hyperbolically indexed nodes for structured queries. We will cite new results on advanced large model performance (your mentioned publications [1,2,3]) in tandem with hyperbolic retrieval layers, illustrating the complementary nature of these approaches.
>
> ### Summary
> We sincerely appreciate your thoughtful feedback and regret that our original discussion did not fully convey how hyperbolic geometry can remain relevant in an LLM-dominated era. Our position paper seeks to underscore the conceptual advantages of negative curvature for deeply hierarchical knowledge, without diminishing the remarkable progress in LLMs. In ongoing work, we plan to provide empirical validations and more comprehensive integration scenarios, illustrating that hyperbolic embeddings function as a strategic layer rather than an outdated alternative in modern health informatics.
>
> We thank you for your frank assessment, which has led us to clarify both the scope and future directions of our proposed pipeline. We hope these clarifications address some of your reservations and inspire continued exploration of how knowledge graph embeddings (hyperbolic or otherwise) can enrich contemporary large-scale healthcare AI systems. We look forward to further discussion and trust these updates will lead to a more favorable evaluation of our manuscript.

---

> > ### Comment · Reviewer_RvWt · 2025-04-03
> >
> > Thank you for the answers. I am not really convinced by the comments on the relevance and "enabling hyperbolic layer", so I will maintain the current score.

---

### Official Review · Reviewer_gHu5 · 2025-03-12

**Significance:** 2
**Argument Clarity:** 2
**Rating:** 2
**Confidence:** 4

**Questions:**

Could you address the issue mentioned in the detailed review?

**Discussion Potential:**

3

**Paper Summary:**

This paper argues for the application of hyperbolic embedding in HKG learning and retrieval. It presented theoretical foundations of, gathered empirical evidence for, addressed doubts against, replacing euclidean embedding with hyperbolic one in this context.


## update after rebuttal
My assessment and score stand. The paper, in its current form, does not sufficiently support its strong position due to the aforementioned omissions and structural weaknesses.

**Position:**

Yes

**Position In Title:**

Yes

**Related Work:**

3

**Strengths And Weaknesses:**

__Strengths__

* Addressed an important domain, healthcare
* Collected and summarized a lot theoretical and empirical evidence for applying hyperbolic embedding


__Weaknesses__

* No discussion of existing empirical evidence against using hyperbolic emb
* Structure of the argument is unclear

__Detailed Review__
* Argument is flawed, suspicious of cherry picking, for omitting a lot of empirical evidence questioning
    * the effectiveness of hyperbolic embedding: the euclidean does just fine, or even better
        * Properly-tuned Euclidean models match or outperform the hyperbolic models on the "most hyperbolic" datasets, (Shedding Light on Problems with Hyperbolic Graph Learning)
        * On popular few-shot datasets, a fixed-radius Euclidean encoder can match and even improve the hyperbolic classification accuracy (Hyperbolic vs Euclidean Embeddings in Few-Shot Learning: Two Sides of the Same Coin)
        * with the right metrics and training objectives, hyperbolic space does not provide any additional benefits compared to Euclidean space for HMTC (Jump To Hyperspace: Comparing Euclidean and Hyperbolic Loss Functions for Hierarchical Multi-Label Text Classification)
    * the scalability
        * Scaling multi-modal hyperbolic models to billions of parameters and increased training complexity has proven particularly difficult. (Hyperbolic Learning with Multimodal Large Language Models)
* The purpose and structure of section 4 is unclear, and it failed to support, even undermine, authors' main statement that we "must" use hyperbolic embedding. It seems to arise many doubts and practical difficulties to adopting hyperbolic representation learning and deployment, but the proposed solutions are not strong enough.
* Literature review missed important discussion about
    - the representation learning and tradeoffs of shallow hyperbolic embedding, such as
        - [Representation Tradeoffs for Hyperbolic Embeddings](https://arxiv.org/pdf/1804.03329.pdf), ICML 2018
				  Frederic Sala, Christopher De Sa, Albert Gu, Christopher Re´
        - Hyperbolic Entailment Cones for Learning Hierarchical Embeddings, ICML 2018
				  Octavian-Eugen Ganea, Gary Bécigneul, Thomas Hofmann
        - Lorentzian Distance Learning for Hyperbolic Representations, ICML 2019
				  Marc T. Law, Renjie Liao, Jake Snell, Richard S. Zemel
        - Shadow Cones: A Generalized Framework for Partial Order Embeddings, ICLR 2024
				  Tao Yu, Toni J.B. Liu, Albert Tseng, Christopher De Sa
        - Tempered Calculus for ML: Application to Hyperbolic Model Embedding, arxiv 2024
				  Richard Nock, Ehsan Amid, Frank Nielsen, Alexander Soen, Manfred K. Warmuth
    - and no mention of the recent hyperbolic neural networks
- The statement in the abs is over-strong that can hardly be supported: "This position paper contends that hyperbolic embeddings **must** become a standard for ..."

**Support:**

3

---

> ### Author Rebuttal · Authors · 2025-03-31
>
> Dear Reviewer gHu5
>
> We appreciate your thorough critique and insightful suggestions. Below, we address the central concerns raised in your detailed review. While this is a position paper rather than a full empirical study, we acknowledge that nuanced perspectives and contrary evidence must be considered to strengthen our stance on hyperbolic embeddings.
>
> ### 1. Acknowledging Empirical Evidence Against Hyperbolic Embeddings
> We recognize that recent works question the purported superiority of hyperbolic methods over carefully tuned Euclidean approaches. For instance:
> (i) Shedding Light on Problems with Hyperbolic Graph Learning shows that, on some “hyperbolic” datasets, standard Euclidean baselines can match or surpass hyperbolic embeddings.
> (ii) Hyperbolic vs Euclidean Embeddings in Few-Shot Learning and Jump To Hyperspace: Comparing Euclidean and Hyperbolic Loss Functions for Hierarchical Multi-Label Text Classification both demonstrate that, with appropriate objectives, Euclidean can yield comparable or even improved performance.
>
> In our ongoing research, we plan to incorporate these findings explicitly in the Related Work section, emphasizing that hyperbolic geometry is not invariably superior and must be evaluated alongside robust Euclidean baselines.
>
> ### 2. Scalability Concerns in Multi-Modal Settings
> You rightly highlight Hyperbolic Learning with Multimodal Large Language Models, which describes the complexity of training large-scale hyperbolic models. We continue to endorse negative curvature for hierarchical data, but we acknowledge the additional computational burdens—particularly for billions of parameters. In future iterations (e.g., Section 4’s expansion), we intend to detail engineering strategies, such as approximate Poincaré-to-Euclidean mappings, to mitigate training and inference costs.
>
> ### 3. Clarifying Section 4’s Purpose and Structure
> Your observation that Section 4 appears to undermine our advocacy of hyperbolic embeddings is valuable. Our intent was to present potential pitfalls (e.g., numeric stability, domain alignment, interpretability) alongside preliminary remedies, yet we now see it can read more like a cautionary note. We propose reorganizing it into two coherent parts:
> (i) Practical Challenges: detailing deployment cost, bridging with existing ANN libraries, etc.
> (ii) Potential Solutions: illustrating curvature tuning, Lorentz-based models, and partial adoption in subgraphs.
>
> This clearer layout should better communicate both the obstacles and concrete ways to address them.
>
> ### 4. Incorporating Missing Literature
> We appreciate your suggestions about key papers and frameworks missing from our initial survey. In subsequent work, we will devote a dedicated subsection to:
> - Representation Tradeoffs for Hyperbolic Embeddings (Sala et al.)
> - Hyperbolic Entailment Cones for Learning Hierarchical Embeddings (Ganea et al.)
> - Lorentzian Distance Learning for Hyperbolic Representations (Law et al.)
> - Shadow Cones: A Generalized Framework for Partial Order Embeddings (Yu et al.)
> - Tempered Calculus for ML: Application to Hyperbolic Model Embedding (Nock et al.)
> - Recent hyperbolic neural network developments (HNN, hyperbolic GNNs).
>
> Highlighting these references will address the dimension-vs.-fidelity tradeoffs, partial-order embedding methods, and advanced optimization strategies.
>
> ### 5. Tone of the Conclusion and Overstatement
> We acknowledge that our phrase “must become a standard” is overly forceful, particularly when multiple lines of research indicate that hyperbolic geometry might not suit every scenario. Our intent was to stimulate discussion on its strong theoretical benefits for hierarchical data; however, we will adjust the wording to “promising alternative,” emphasizing that hyperbolic embeddings are especially helpful for deeply nested ontologies rather than a universal solution.
>
> ### Summary
> We sincerely appreciate your thorough review and remain committed to addressing all of your concerns. Specifically, we will explicitly reference Euclidean competitiveness to account for contradictory evidence, expand our discussion of scalability and complexity -- especially for hyperbolic models in multimodal tasks -- and reorganize Section 4 to systematically present challenges before proposing solutions. We will also integrate key works on partial orders and Lorentzian methods. Finally, we plan to moderate our claims about the necessity of hyperbolic embeddings by acknowledging their context-dependent applicability. We hope these refinements will favorably address your feedback and lead to a higher assessment of our on-going work.
> Again, we extend our sincere gratitude for your invaluable insights.

---

> > ### Comment · Reviewer_gHu5 · 2025-04-09
> >
> > While I appreciate the authors' engagement with the feedback, the rebuttal primarily outlines promised changes for future iterations or ongoing work ("plan to incorporate," "intend to detail," "propose reorganizing," "will devote," "will adjust").
> >
> > As the review process assesses the paper as submitted, and the rebuttal confirms these issues will only be potentially addressed in future work, the fundamental weaknesses pointed out in the initial review remain.
> >
> > Therefore, my assessment and score stand. The paper, in its current form, does not sufficiently support its strong position due to the aforementioned omissions and structural weaknesses.

---

### Official Review · Reviewer_iVAR · 2025-03-16

**Significance:** 3
**Argument Clarity:** 3
**Rating:** 3
**Confidence:** 5

**Questions:**

•	In the Hyperbolic HKG pipeline, the knowledge entities are embedded into hyperbolic space, how are they applied to LLM queries? Figure 1 indicates that the LLM invokes the Vector DB for query retrieval. However, the LLM itself employs Euclidean space embedding for token training and learning, and the output results are natural text sequences; whereas hyperbolic embedding represents standardized knowledge entities. So, how does the entire pipeline establish the association from natural text sequences to standardized knowledge entities? If the encoder for hyperbolic embedding is reused, how does the pipeline achieve the transition from Euclidean space embedding to hyperbolic space embedding?

•	In the Vector DB query retrieval section, the similarity calculation is not clearly explained. Is it based on hyperbolic space geodesic distance, or does it still use Euclidean space cosine distance? Furthermore, in hyperbolic space, the entailment relationship can more intuitively depict the hierarchy among knowledge entities and measure the semantic correlation between entities. Has this aspect been considered?

•	In the hyperbolic embedding section, the paper introduces the hyperbolic manifold of the Poincaré disk, but it exhibits numerical instability during modeling and learning. In recent years, many studies have adopted the Lorentzian manifold for embedding learning. It is suggested that the authors include more discussions on hyperbolic embedding.

•	In medical knowledge graphs, there exist complex relationships among various entities, such as those between major diseases and subgroups, relationships within the same subgroup of diseases, causal relationships between diseases, and relationships between diseases and symptoms. From the perspective of semantic relationships, these can be viewed as hyponymy, synonymy, entailment, etc. Current hyperbolic embedding primarily focus on modeling entailment relationships. In the face of the challenges posed by knowledge graphs with complex semantic relationships, have the limitations and opportunities of hyperbolic modeling been considered and discussed?

•	In recent years, hyperbolic modelling has also been used in the modeling of medical data, such as image-text representation learning and knowledge enhancement. Compared to these works, have the differences and innovations of this work been discussed?"

**Discussion Potential:**

3

**Paper Summary:**

This paper advocates for hyperbolic embeddings as the standard for modeling and retrieving hierarchical health knowledge graphs in LLMs, arguing they better capture medical data's hierarchies and patterns. It presents theoretical and practical strategies, supported by appendices.

**Position:**

Yes

**Position In Title:**

Yes

**Related Work:**

2

**Strengths And Weaknesses:**

Advantages:

1. Hyperbolic geometry holds an advantage when modeling data with hierarchical relationships. Combining health knowledge graphs to enhance the understanding and expressive capabilities of Large Language Models (LLMs) is novel and logically sound.
2. The article not only presents this viewpoint but also provides theoretical support and practical implementation strategies.
3. It elaborates on the significance of hyperbolic embeddings in detail and offers in-depth reflections from various perspectives in the discussion section.

Disadvantages:

1.  For the primary application scenario of enhancing LLMs, within the proposed systematic framework, the integration of knowledge graphs with LLMs is not clearly explained.
2. Regarding the embedding of complex knowledge graphs, the disadvantages of hyperbolic modeling are not discussed in detail. It remains unclear whether the current solutions are suitable for complex medical knowledge graphs.
3. Related work is lacking. While the article revolves around medical data modeling, the related work section does not cover explorations into hyperbolic embeddings specifically for medical data modeling.

**Support:**

3

---

> ### Author Rebuttal · Authors · 2025-03-31
>
> Dear Reviewer iVAR
>
> We greatly appreciate your detailed feedback and questions. Below, we provide point-by-point responses to issues raised in the main text and Appendices, further clarifying and expanding upon the key elements of our proposed "Hyperbolic Health Knowledge Graphs (HKG) Pipeline."
>
> ### Q1. Associating Natural Language Sequences with Standardized Hyperbolic Entities
> A1: We preserve the LLM’s Euclidean token embeddings internally but introduce a projection step that maps these embeddings into hyperbolic coordinates when querying the Vector DB. This approach (detailed in Section 3.2) avoids invasive modifications to the LLM’s architecture while harnessing negative curvature for retrieval. In principle, one could attach a final Riemannian projection layer to the LLM encoder itself, but our current design places this module externally to minimize disruptions and maintain compatibility with standard LLMs.
>
> ### Q2. Similarity Computation in the Vector DB and Entailment Relations
> A2: Our proposal generally relies on hyperbolic distances (e.g., Poincaré or Lorentz geodesics) for retrieval. As noted in Sections 2.2 and 3.3, this naturally preserves hierarchical fidelity and encodes “is–a” structures more compactly than Euclidean alternatives. For real-world scaling with existing ANN systems (like FAISS or HNSW), we also discuss approximate mappings that project hyperbolic vectors to a Euclidean surrogate. This compromise retains most hierarchical properties while capitalizing on mature indexing tools. Additionally, hyperbolic embeddings can capture entailment relations, particularly where parent–child or partial-order structures exist (Appendix B).
>
> ### Q3. Numerical Instability of the Poincaré Disk and the Use of the Lorentz Model
> A3: Although Poincaré disk embeddings can exhibit numerical instability as points approach the boundary (Section 2.3; Appendix A), we note that mathematically equivalent Lorentz-space (hyperboloid) embeddings often improve stability and gradient flow for large datasets. Our pipeline can incorporate Lorentz embeddings with minimal changes to the overall design. Ongoing work includes benchmarking both parameterizations under realistic multi-level medical ontologies to assess which model handles boundary effects more robustly.
>
> ### Q4. Complex Semantic Relationships in Medical KGs
> A4: While hyperbolic models excel at parent–child or hierarchical entailment edges, real-world medical KGs also exhibit lateral, causal, and synonym relationships (Section 4.2). We propose multi-relation extensions (such as using different geodesic transformations per relation type) and hyperbolic GNN layers that combine radial depth for hierarchy with tangential directions for cross-branch links. These enhancements can unify synonyms and causal loops in a single negative-curvature space, though additional modeling is required beyond a purely tree-based approach.
>
> ### Q5. Distinctions from Existing Medical Hyperbolic Modeling (Image–Text or Knowledge Enhancement)
> A5: Our work delivers an integrated “Hyperbolic HKG Pipeline” rather than a single-task demonstration. We focus on (i) curvature calibration for multi-scale medical ontologies (Appendix A), (ii) end-to-end integration with LLM-driven queries, and (iii) modular deployment, including vector indexing and clinician-facing interpretations (Appendix D). Previous studies often address narrower tasks (e.g., hyperbolic embeddings for image–text retrieval), whereas we emphasize a production-ready platform that unifies ingestion, embedding, retrieval, and LLM inference across large-scale clinical taxonomies.
>
> ### Summary
> We sincerely appreciate your thoughtful evaluation of our work. By mapping hierarchical and small-world relationships into a hyperbolic space, our pipeline aim to achieve superior compression and clearer boundary distinctions compared to Euclidean models. Through a lightweight projection layer, we align the LLM’s textual representation with the hyperbolic domain, enabling more faithful retrieval of deeply nested medical concepts. For large-scale deployments, approximate projection schemes support rapid ANN queries, while Lorentz-based parameterizations address potential numerical instabilities.
> Our approach also extends to multi-relation embeddings, ensuring broader coverage of real-world causal and lateral links in HKGs. Future directions include federated hyperbolic learning for privacy-preserving data sharing and refined interpretability to bolster clinical trust. We hope these enhancements address your concerns and earn a more favorable evaluation of our manuscript.
> Again, we thank you for your insightful review comments.

---

### Decision · Program_Chairs · 2025-04-27

**Decision:**

Reject

**Comment:**

A major concern from reviewer is that this position paper fails to show that hyperbolic embedding is really better than the
Euclidean embedding for real-world biomedical applications. In fact, this point is quite important. If there were no
sufficient evidence to support the superiority of hyperbolic embedding, then this direction is questionable.